# Immunogenicity and Efficacy of Combined mRNA Vaccine Against Influenza and SARS-CoV-2 in Mice Animal Models

**DOI:** 10.3390/vaccines12111206

**Published:** 2024-10-24

**Authors:** Elena P. Mazunina, Vladimir A. Gushchin, Evgeniia N. Bykonia, Denis A. Kleymenov, Andrei E. Siniavin, Sofia R. Kozlova, Evgenya A. Mukasheva, Elena V. Shidlovskaya, Nadezhda A. Kuznetsova, Evgeny V. Usachev, Vladimir I. Zlobin, Elena I. Burtseva, Roman A. Ivanov, Denis Y. Logunov, Alexander L. Gintsburg

**Affiliations:** 1N. F. Gamaleya Federal Research Center for Epidemiology & Microbiology, Ministry of Health, Moscow 123098, Russia; evgeniya_bikonya@mail.ru (E.N.B.); mne10000let@yandex.ru (D.A.K.); andreysi93@ya.ru (A.E.S.); sofya_dadashyan@mail.ru (S.R.K.); mukasheva_evgeniya@mail.ru (E.A.M.); lenitsa@gmail.com (E.V.S.); nadyakuznetsova0@yandex.ru (N.A.K.); evgenyvusachev@gmail.com (E.V.U.); vizlobin@mail.ru (V.I.Z.); elena-burtseva@yandex.ru (E.I.B.); ldenisy@gmail.com (D.Y.L.); gintsburg@gamaleya.org (A.L.G.); 2Department of Virology, Lomonosov Moscow State University, Moscow 119234, Russia; 3Department of Medical Genetics, I.M. Sechenov First Moscow State Medical University (Sechenovskiy University), Ministry of Health, Moscow 119991, Russia; 4Shemyakin-Ovchinnikov Institute of Bioorganic Chemistry, Russian Academy of Sciences, Moscow 117997, Russia; 5Translational Medicine Research Center, Sirius University of Science and Technology, Sochi 354340, Russia; ivanov.ra@talantiuspeh.ru; 6Infectiology Department, I. M. Sechenov First Moscow State Medical University (Sechenovskiy University), Ministry of Health, Moscow 119991, Russia

**Keywords:** mRNA vaccine, seasonal influenza, SARS-CoV-2, combined vaccine, immunogenicity, efficacy, immune response

## Abstract

**Background.** The combined or multivalent vaccines are actively used in pediatric practice and offer a series of advantages, including a reduced number of injections and visits to the doctor, simplicity of the vaccination schedule and minimization of side effects, easier vaccine monitoring and storage, and lower vaccination costs. The practice of widespread use of the combined vaccines has shown the potential to increase vaccination coverage against single infections. The mRNA platform has been shown to be effective against the COVID-19 pandemic and enables the development of combined vaccines. There are currently no mRNA-based combined vaccines approved for use in humans. Some studies have shown that different mRNA components in a vaccine can interact to increase or decrease the immunogenicity and efficacy of the combined vaccine. **Objectives.** In the present study, we investigated the possibility of combining the mRNA vaccines, encoding seasonal influenza and SARS-CoV-2 antigens. In our previous works, both vaccine candidates have shown excellent immunogenicity and efficacy profiles in mice. **Methods.** The mRNA-LNPs were prepared by microfluidic mixing, immunogenicity in mice was assessed by hemagglutination inhibition assay, enzyme-linked immunoassay and virus neutralization assay. Immunological efficacy was assessed in a mouse viral challenge model. **Results.** In this work, we demonstrated that the individual mRNA components of the combined vaccine did not affect the immunogenicity level of each other. The combined vaccine demonstrated excellent protective efficacy, providing a 100% survival rate when mice were infected with the H1N1 influenza virus and reducing the viral load in the lungs. Four days after the challenge with SARS-CoV-2 EG.5.1.1., no viable virus and low levels of detectable viral RNA were observed in the lungs of vaccinated mice. **Conclusions.** The combination does not lead to mutual interference between the individual vaccines. We believe that such a combined mRNA-based vaccine could be a good alternative to separated human vaccinations for the prevention of COVID-19 and influenza.

## 1. Introduction

In the European region of Eurasia alone, influenza viruses infect up to 50 million people each year and cause at least 15–70 thousand deaths [1]. At the same time, long-term observations show that the increase in influenza morbidity in the human population is seasonal, necessitating the use of widespread vaccine prophylaxis before each season. Despite the World Health Organization (WHO) ending the global emergency status for COVID-19, SARS-CoV-2 virus continues to circulate in the human population. In the last 6 months of 2024, approximately 2.4 million new cases of COVID-19 were detected worldwide, resulting in 33.5 thousand deaths [2]. Like influenza viruses, SARS-CoV-2 are subject to rapid evolution, resulting in the emergence of new variants of concern to the WHO [3].

In the case of the SARS-CoV-2 virus, the increase in incidence is not so closely related to the season, although there is a tendency towards seasonality [4]. The increase in incidence appears to be linked to the emergence of new virus variants that evade the pre-existing immunity developed in the human population. New waves of increased incidence are observed more often than once a year, indicating the need for regular revaccination. Thus, in the early stages after the introduction of vaccines, the practice of regular booster immunizations was applied [5,6,7]. Revaccination with an original antigenic formulation based on the Spike (S) protein of the wild-type (Wuhan-Hu-1 strain) was shown to provide protection against Delta and early Omicron variants [8,9,10]. After two years of widespread immunization with the original Wuhan vaccine antigen composition, a change in vaccine composition to include the Omicron variant’s antigen was required. The updated BA.1, BA.4.5, and XBB.1.5 vaccines have been in use for the past few years [11]. Currently, the WHO recommends use of a monovalent JN.1 lineage as the antigen in future formulations of COVID-19 vaccines [12]. The absence of, for many people, a clear logic to the regular vaccine update has led to a sharp decline in their use [13]. Statistics show that while the vast majority of people (60–90%) in developed countries have been vaccinated with the original antigenic formulation, far fewer people are vaccinated with the updated formulation [14]. Even in high-risk groups, only a small number of people are vaccinated [15,16]. This poses a risk of reduced population immunity and, as a consequence, a significant increase in COVID-19 morbidity and mortality in the near future. The solution to this problem may be to switch to the routine administration of a vaccine against COVID-19 with an updated antigenic composition. The easiest way to do this is to combine it with the seasonal influenza vaccination, as it is the only vaccine that is administered annually to adults, children, and high-risk individuals with truly widespread coverage [17].

Several mRNA-based combination products are currently undergoing nonclinical studies that are showing good immunogenicity and efficacy profiles in mice [18,19]. While the world leaders in mRNA vaccine production, Moderna and Pfizer-BNT, are simultaneously conducting clinical trials of combined mRNA vaccines against influenza and coronaviruses (NCT05375838, NCT06097273, and NCT05596734, respectively), the results of the preclinical studies of these combination products have not yet been published. Based on our recently studied mRNA platform, we combined two previously described mRNA-vaccines against influenza and SARS-CoV-2 for the simultaneous specific prophylaxis of seasonal influenza and COVID-19 [20,21]. In this work, we examined how they mutually influence each other in terms of their immunogenicity and efficacy and assessed them to appreciate the benefits of this multivalent mRNA combination.

## 2. Materials and Methods

### 2.1. mRNA Production and mRNA-LNP Assembly

The coding regions of influenza hemagglutinins (HAs) and Spike of SARS-CoV-2 were cloned into the pJAZZ-OK-based linear bacterial plasmids (Lucigen) separately, as described earlier [22]. The Sanger sequencing was used for the confirmation of the designed sequences. Plasmid accumulation and isolation and in vitro mRNA transcription were performed, as previously described [21].

mRNA-LNP assembly was carried out using the microfluidic mixing of lipid components (ionizable lipid:DSPC:cholesterol:PEG-lipid) and mRNA in acidification buffer of 10 mM sodium citrate (pH 3.0). The mRNA encapsulation efficiency and concentration were determined by SYBR Green dye (SYBR Green I, Lumiprobe, Moscow, Russia) followed by fluorescence measurement, as described earlier [20]. 

### 2.2. Viruses

Influenza virus was propagated in embryonated chicken eggs or MDCK cells using conventional techniques, as previously described [23,24]. Determination of virus titer was carried out by endpoint dilution assay in MDCK cells, as previously described [24]. SARS-CoV-2 viruses (strain EG.5.1.1; GISAID EPI_ISL_18543695, XBB.1; and GISAID EPI_ISL_16053000) were amplified and titrated on Vero E6 cells [25]. The stocks of SARS-CoV-2 used in the experiments had undergone three passages. Viral titers were determined in confluent Vero E6 cells as TCID_50_ in 96-well microtiter plates by endpoint dilutions assay. Virus-related work was carried out under high-containment Biosafety Level 3 (BSL-3) conditions.

### 2.3. Animal Studies

Female 4–5-week-old mice were used to study the immunogenicity and immunological efficacy of candidate vaccines in challenge experiments. Animals were purchased from Stolbovaya nursery (Research Center for Biomedical Technologies of FMBA; Chekhov, Russia). Animal experiments were performed under Protocol #41 from 6 April 2023, as approved by the Biomedical Ethics Committee of the Federal Research Centre of Epidemiology and Microbiology named after Honorary Academician N.F. Gamaleya. Animal care and experimental work were carried out in accordance with the Directive 2010/63/EU, FELASA recommendations [26], and the inter-state standard of “GLP” [27].

### 2.4. Hemagglutinin Inhibition (HAI) Assay

Immunogenicity of influenza components in animal experiments was estimated by hemagglutinin inhibition assay, according to the WHO-based HAI protocol, as previously described [21]. Antigens of influenza virus for HAI test (A/Wisconsin/588/2019, A/Darwin/9/2021, B/Austria/1359417/2021) were purchased from LLC “Company for the production of diagnostic drugs” (St-Petersburg, Russia) or propagated in embryonated chicken eggs (B/Washington/02/2019).

### 2.5. Virus Neutralization Assay

The estimation of neutralization activity of sera was carried out with neutralization assay [20]. Briefly, sera from vaccinated animals were serially diluted in complete DMEM supplemented with 2% fetal bovine serum (FBS) and mixed with 1000 TCID_50_/mL of the corresponding SARS-CoV-2 variant and incubated at 37 °C for 1 h. The serum–virus mixtures were then incubated with monolayer of Vero E6 cells for 96 h. The cytopathic effect (CPE) was assessed visually. The highest serum dilution that inhibited the development of CPE by at least 80% was identified as the neutralization titer. Amplification and titration of the SARS-CoV-2 viruses used in this work were carried out as previously described [28,29]. The virus titer was determined using the Reed and Muench method and expressed as TCID50/mL (50% tissue culture infectious dose).

### 2.6. ELISA 

ELISA was carried out as described previously [21], with some differences. Plates (96-well; Servicebio, Wuhan, China) were coated with 100 μL of recombinant RBD of SARS-CoV-2 virus Omicron strain XBB.1 (EVV00331; AntibodySystem, Schiltigheim, France) in PBS at 1 μg/mL and incubated overnight at 4 °C. Four-fold serial dilutions of serum samples were used in the analysis with the initial sample dilution 1:10.

### 2.7. Viral Challenge 

The lethal influenza virus infection model was performed in 4–5-week-old females of B.6Cg-Tg(K18-ACE2) mice, using a mouse adapted (m.a.) influenza virus A/Victoria/2570/2019 (H1N1)pmd09, as previously described [21]. Mice were intranasally infected with 50 µL of virus suspension under Zoletil-Xylazine anesthesia. Animals were monitored daily for clinical signs of infection (weight loss and survival) for 15 days after challenge. Time of death was defined as the time at which a mouse was found dead or was euthanized.

The COVID-19 infection caused by SARS-CoV-2 was performed on 4–5-week-old females of B.6Cg-Tg(K18-ACE2) mice, as described earlier [20]. Mice were infected intranasally with 50 µL of virus suspension (SARS-CoV-2 virus strain EG.5.1.1) under Zoletil-Xylazine anesthesia. Endpoints were on 4 and 7 days after infection, with subsequent lung isolation and blood collection. 

### 2.8. Quantification of Virus in Infected Lungs from Mice 

Lungs were harvested from mice 3 days after influenza infection or 4 and 7 days after SARS-CoV-2 infection. After harvest, the lungs were weighed and then homogenized in sterile DMEM supplemented with gentamicin to produce a 20% lung-in-medium solution. Total RNA was extracted from lung homogenates using ExtractRNA reagent (Eurogen, Moscow, Russia) according to the manufacturer’s instructions. Amplification and quantification of Influenza A and SARS-CoV-2 virus RNA was performed using a one-step RT-qPCR technique, as described previously [21]. The primers and probes were designed to target the gene coding M (matrix protein) of Influenza A virus and NSP1 (leader protein) of SARS-CoV-2. The oligonucleotides were as follows: for Influenza A:○forward primer—5′- ATG GAG TGG CTA AAG ACA AGA C -3′, ○reverse primer—5′- GCA TTT TGG ACA AAG CGT CTA -3′, ○probe 5′-FAM—TCC TCG CTC ACT GGG CAC GGT -BHQ1-3′ for SARS-CoV-2 ○forward primer—5′- GTA CGT GGC TTT GGA GAC TC -3′, ○reverse primer—5′- ACT AAG CCA CAA GTG CCA TC -3′, ○probe 5′-Cy5- AGG AGG TCT TAT CAG AGG CAC GTC A -BHQ2-3′. 

Amplification was performed using a real-time CFX96 Touch instrument (Bio-Rad, Hercules, CA, USA). The conditions of the one-step RT-qPCR reaction were as follows: 50 °C for 15 min and 95 °C for 5 min, followed by 45 cycles of 95 °C for 10 s and 55 °C for 1 min. The number of copies of viral RNA was calculated using a standard curve generated by amplification of a plasmid cloned DNA template containing the amplified fragment.

### 2.9. Blood Sample Collection and Hematology

For hematological analysis, K18-hACE2 mice vaccinated with the mRNA-5 and PBS were anaesthetized with Zoletil-Xylazine and 0.5 mL of blood was collected from the cranial vena cava. A Smart V5 Vet (URIT Medical Electronic Group Co., Shenzhen, China) automated hematology analyzer was used to perform the assay according to the manufacturer’s protocol. The numbers of white blood cells (WBC), lymphocytes (LYM), neutrophils (NEU), monocytes (MON), eosinophils (EOS), normoblasts (NRBC), and hemoglobin (HGB) were analyzed.

### 2.10. Statistical Analysis

Data were analyzed using GraphPad Prism software version 9.5.0. Data of immunogenicity, viral load, neutralization titer, and hematology were analyzed using a Mann–Whitney test. Survival data were compared using the Mantel–Cox long-rank test with Dunn’s multiple comparison test and weight loss was compared using Tukey’s multiple comparison test.

## 3. Results

### 3.1. Antigen Selection, pDNA Design, and Candidate Vaccine Production

The antigenic composition of the influenza components of the combined vaccine was based on the WHO recommendation for a trivalent seasonal influenza vaccine in 2022–2023. The B/Massachusetts/02/12 strain was chosen for the fourth influenza component (B Yamagata lineage). The haemagglutinin gene sequences of the selected influenza strains were obtained from GenBank, then codon optimized and synthesized. The antigen of the XBB.1 sub-lineage of the Omicron variant, whose descendent lineages became globally dominant in the first half of 2023, was selected as the coronavirus component of the combined vaccine. The spike protein gene was codon optimized and synthesized. The vector used for DNA template assembly was the linear bacterial plasmid pJAZZ-OK (previously described [22]), which, according to our data, ensures stability of the length of the poly(A) region of the mRNA (data have not been published yet). All sequences were verified for regulatory parts and CDS by Sanger sequencing. Multivalent vaccines for immunization of the animals in this study were obtained by separate formulation of each mRNA and mixing of mRNA-LNPs prior to vaccination.

### 3.2. Immunogenicity of a Combined Multivalent mRNA Vaccine in Mice

To assess the immunogenicity of the combined mRNA vaccine, female BALB/c mice were immunized with a formulation (mRNA-5) containing 5 µg each of mRNAs encoding hemagglutinins of the influenza viruses and the spike protein of SARS-CoV-2 XBB.1 (Table 1).

The total mRNA content in one dose was 25 µg. The vaccine was administered according to a two-dose intramuscular regimen, with a 21-day interval between doses. As control vaccines, the following were used: an approved split inactivated influenza vaccine (SIIV, 3 µg of each of the 4 antigens per mouse), an mRNA-XBB encoding the spike protein of SARS-CoV-2 XBB.1 (5 µg/mouse), a quadrivalent mRNA influenza vaccine (mRNA-4) encoding the same hemagglutinins as the combined vaccine (5 µg of each mRNA/mouse), the updated vector-based COVID-19 vaccine Sputnik V XBB.1 (1/5 of the human dose), and sterile phosphate-buffered saline (PBS) as placebo.

The rationale for the dose of the combined vaccine is based on the results of immunogenicity and efficacy studies of mRNA vaccines in previous research [20,21]. The dose of SIIV for mice, equal to 1/5 of the human dose, was chosen because, in a previous study, a lower dose (1/10 of the human dose) showed a very low level of immunogenicity for all components. The dose of Sputnik V (XBB.1), equal to 1/5 of the human dose (2 × 10^10^ viral particles), is justified by the results of earlier preclinical studies of the Sputnik V vaccine [30]. Blood samples were collected from the mice twice—14 and 39 days after the first vaccination.

#### 3.2.1. Immunogenicity of the Influenza Component

The immunogenicity assessment of the influenza component was conducted using the hemagglutination inhibition (HAI) assay with various antigens included in the mRNA formulations and/or the inactivated split vaccine (A/Wisconsin/588/2019 (H1N1)pmd09, A/Darwin/9/2021 (H3N2), B/Austria/1359417/2021 (B/Victoria lineage), B/Massachusetts/02/12 (B/Yamagata lineage), and B/Phuket/3073/2013 (B/Yamagata lineage)). This analysis was performed on the sera of mice from the mRNA-5, mRNA-4, SIIV, and PBS groups according to the design showed on Figure 1a. The results of the inter-group comparison are presented in Figure 1b–f.

A comparison of the immunogenicity levels between combined vaccine (mRNA-5), quadrivalent mRNA vaccine (mRNA-4), and SIIV showed that the fifth component in the mRNA-5 vaccine (mRNA encoding the SARS-CoV-2 S protein) does not affect the immunogenicity of the influenza components. Regarding the intensity of the immune response to the influenza components, we observed expected results that replicate previously published data for the trivalent mRNA vaccine [21]. The boost dose effect manifested as a significant increase in antibody levels on the 18th day after the second immunization (Figure 1b–f). In particular, after the second immunization with the mRNA-5 vaccine, serum antibody titers in the HAI assay increased by at least 10-fold on average for all antigens tested.

Consistent with our previous work, the immune response induced by the SIIV against the influenza A H3N2 virus exhibited considerable variability, with most HAI titer at the lower detection limit. On day 39, only a slight increase in immune response to the second dose was observed, resulting in statistically insignificant differences in the mean HAI titer after the first and second doses of SIIV (Figure 1c).

The mRNA vaccine strain B/Massachusetts/02/2012 is heterologous compared to the SIIV strain B/Phuket/3073/2013. Assuming that this heterology could explain the low immune response to this antigen in mice that received the SIIV (Figure 1e), we conducted an additional HAI analysis using the B/Phuket/3073/2013 virus antigen (Figure 1f). The results still demonstrated the significant advantages of the mRNA-5 and mRNA-4 vaccines over SIIV (differences of 25-fold and 20-fold, respectively). This underscores our previously stated assertion that split inactivated seasonal vaccines provide a weak cross-immune response to heterologous strains within influenza virus subtypes at both investigated dosages—1/5 and 1/10 of the human dose per mouse [21]. In contrast, the mRNA vaccine ensures a consistently higher antibody titer at a dosage of 5 µg of each mRNA per mouse.

#### 3.2.2. Immunogenicity of the SARS-CoV-2 S-Glycoprotein

The immunogenicity of the pentavalent mRNA-based vaccine against coronavirus was evaluated by measuring the levels of binding antibodies (after the first and second immunizations) and the virus-neutralizing activity (VNA) of serum from immunized mice collected on day 39 of the study. To assess the level of binding antibodies generated in response to the vaccination with the mRNA-5 combined vaccine and control vaccines, the sera from immunized animals were analyzed by ELISA. The results are presented as titration curves of the sera and comparisons of the area under these curves (AUC) (Figure 2a–e). The amount of serum collected from the tail vein on day 14 after the first immunization was insufficient for VNA testing. VNA titers of the serum from vaccinated mice were evaluated against two SARS-CoV-2 variants (XBB.1 and EG.5.1), with the results being presented in Figure 2e. A statistical comparison of VNA titers between the monovalent and combined mRNA vaccines revealed no significant differences, indicating that the influenza components in the vaccine formulation did not affect the immunogenicity of the coronavirus component. No differences in VNA titers against the two coronavirus strains were detected, suggesting no reduction in the neutralizing activity of antibodies generated by the mRNA vaccine against the more recent sub-lineage of the Omicron variant—EG.5.1.

Statistical comparison of AUC values after the first immunization showed significant differences between the investigated vaccines, control vaccines, and the PBS group (Figure 2a). However, analysis of AUC after the second immunization revealed significant differences in immunogenicity between the mRNA vaccines (mRNA-5 and mRNA-XBB) and Sputnik V (XBB.1) and the PBS (Figure 2b). Nonetheless, no significant differences were found between the monovalent mRNA formulation (mRNA-XBB) and the combined vaccine (mRNA-5) in this analysis.

### 3.3. Efficacy of the Combined mRNA Vaccine in Mice in Experiments on Influenza and Coronavirus Challenge

The ability of the investigated combined vaccine to protect immunized animals from lethal outcomes was tested in experiments involving infection with the influenza A virus strain A/Victoria/2570/2019 (H1N1)pmd09 (m.a.), which is adapted for use in a mouse infection model. Additionally, the ability of the combined mRNA vaccine to protect mice from high viral loads in the lungs was studied by infecting them with the SARS-CoV-2 strain EG.5.1. For this purpose, 4–5-week-old male mice of the transgenic B.6Cg-Tg(K18-ACE2) line were vaccinated according to the two-dose schedule shown in Figure 3a.

One dose per mouse of the mRNA-5 vaccine consisted of 5 µg of each of the mRNA encoding proteins of influenza and SARS-CoV-2 viruses. The animals in the control group were immunized with PBS. Twenty-one days after the full vaccination regimen, the mice were separately infected with influenza and coronavirus (Figure 3a). The infectious dose of the influenza A/Victoria/2570/2019 (H1N1)pmd09 virus m.a. was 20 LD_50_ per animal. The viral dose per K18-ACE2 mouse was determined in a preliminary experiment. After the influenza infection, survival and weight loss dynamics in the mice were monitored. Three animals from every group were dissected to analyze the viral load in the lungs 3 days post infection. Visually, during dissection, necrotic foci in the lung tissue were observed in the control group (PBS) mice, while the lungs of animals vaccinated with the mRNA-5 vaccine showed no obvious necrotic foci and remained clear (Figure 3d). In the control group of mice vaccinated with PBS, 100% mortality was observed by the 7th day after infection (Figure 3b). At the same time, the group vaccinated with the mRNA-5 vaccine demonstrated 100% survival up to the end of the experiment. The mRNA-5 group showed an average weight loss of 10% during the first 4 days after infection, after which the weight of the mice had recovered by day 9 and continued to increase until the end of the observation period (Figure 3c). As shown by the results of the viral load analysis in the lungs on day 3 post infection, vaccination of mice with the combined mRNA vaccine promotes significantly lower influenza virus content in the lungs (Figure 3e,f).

The mouse model of coronavirus infection involved intranasal infection of vaccinated B.6Cg-Tg(K18-ACE2) mice with the SARS-CoV-2 EG.5.1 Omicron lineage virus (10^6^ TCID_50_ per mouse). Four and seven days after infection, the animals (n = 4 per group) were euthanized to harvest the lungs and collect blood samples. According to our observations, confirmed by various studies [28,31,32], coronavirus strains belonging to the Omicron lineage do not exhibit 100% lethality in the B.6Cg-Tg(K18-ACE2) mouse model, which is why we chose this design to investigate the efficacy of the mRNA vaccine. In the lung homogenates obtained, viral load and TCID_50_ were determined, with the results presented in Figure 3g,h. On the seventh day, no viable virus was detected in the lungs of either the control or mRNA-5 vaccinated animals during TCID50 analysis. However, the lungs of the control group animals showed visual signs of tissue necrosis throughout the organ, which is likely to be related to the absence of viable virus in the TCID50 analysis. Additionally, hematological analysis was performed for collected blood samples (Figure 4). 

Significant differences were found when comparing the values of two parameters between the PBS and mRNA-5 vaccinated mouse groups: the relative number of eosinophils (EOS) and nucleated red blood cells (NRBC, Figure 4).

## 4. Discussion

Based on the results of the worldwide vaccination program against SARS-CoV-2, it is evident that the antigenic composition of the coronavirus vaccine should be updated according to the study of several key properties of new WHO-designated Variants of Concern (VOC) [33]. Previously, we have examined early indicators of the need to change the antigenic composition of SARS-CoV-2 vaccines, including mutations in the S protein, antigenic characteristics, the ability of the current vaccine to reduce infection and transmission, and the range of clinical manifestations, with the most important being cross-neutralizing activity against circulating and new VOC variants [4].

There are several arguments in favor of multi-pathogen combining vaccines: reducing the need for multiple injections, simplifying the immunization schedule and minimizing side effects, reducing the number of doctor visits, simplifying vaccine monitoring and storage, lowering vaccination costs, timely vaccination according to the schedule, and reducing the risk of needlestick injuries for healthcare workers [34]. According to various studies, the use of combined vaccines has been associated with an increase in vaccination coverage [35,36,37]. According to data from Rospotrebnadzor, flu vaccination coverage in 2023 in Russia reached 76.5 million people [38], while both components of the coronavirus vaccine were administered to 79.7 million people (54% of the population) over 2.5 years (from December 2020 to June 2023) [39]. In the Russian Federation, starting in 2024, despite the availability of the updated Sputnik V vaccine [40], vaccination is recommended only for high-risk groups, leading to a sharp decline in vaccine use. Similar trends are observed in other countries [15,16]. It is possible that the introduction of a combined vaccine with a flu component, with a proven safety and efficacy profile, will help increase vaccination coverage against coronavirus, thereby providing the population with protection against COVID-19.

In our two previous publications, we reported on the high immunogenicity and cross-specific activity of candidate vaccines against influenza and coronavirus separately [20,21]. In those studies, we showed that the in vitro synthesized mRNAs have functional activity as transfected cells and produced the protein encoded in these mRNAs. In the current study, we demonstrated that combining these two vaccines (mRNA encoding HAs of four influenza viruses and the spike protein of SARS-CoV-2) does not reduce the immunogenic effect caused by the individual components after two-dose vaccination of mice. For the flu components, the immunogenicity levels in HAI analysis reproduced the previously published results. Importantly, no significant differences in HAI titers were found between the quadrivalent vaccine (mRNA-4) and the pentavalent combined one (mRNA-5), indicating no influence of the coronavirus component in the mRNA-5 composition. As in the previous study, the immunogenicity level of the control vaccine—a split inactivated influenza vaccine—was significantly lower against all four influenza virus antigens. Our results are consistent with those of researchers from China, who, while studying a combined mRNA vaccine with a total dose of 30 µg in mice, did not find there to be mutual influence of the components on their immunogenicity levels [18]. Meanwhile, a combined intranasal vaccine based on chimpanzee adenovirus AdC68-HATRBD showed lower immunogenicity in HAI, with the GMT not exceeding 1:46 after the prime dose and 1:80 after the second immunization against the H1N1 influenza virus [41].

Regarding the immunogenicity of the coronavirus component of the vaccine, it is worth noting that the combined mRNA vaccine demonstrates high immunogenicity, indistinguishable from that of the monovalent vaccine after second vaccination. However, after first vaccination, a significant excess of immunogenicity level in mice that received monovalent mRNA vaccine compared to multivalent was observed. We can speculate that the effect obtained is related to the specificity of the primary immune response development, which is based on the ability of local professional antigen-presenting cells to take up antigen and present it to T-cells and non-specific effector cells. Although mice in the mRNA-5 and mRNA-XBB groups received the same amount of mRNA encoding the spike antigen, the mice in the multivalent group received a total of 5 times more mRNA (25 µg of total mRNA per mouse). Therefore, mice from these groups received different amounts of antigen load per conditionally equal number of regional DCs, which could be the result of a decrease in the level of binding antibodies to the coronavirus adhesion protein. A similar effect of the mutual influence of vaccine mRNA components on the level of immunogenicity has been previously demonstrated [20,42]. In two other studies, vaccination with the combination of 4 to 20 mRNAs encoding influenza antigens did not result in reduced immunogenicity compared to monovalent preparations [43,44]. However, the dose of monovalent mRNAs and combinations per mouse, the route of administration, and the time of sera collection were different in our study.

At the time of creating the coronavirus mRNA component, the dominant Omicron variants were the XBB sub-lineages (XBB.1.5, XBB.1.16, and XBB.1.9.1). In our laboratory, a pDNA construct encoding the codon-optimized sequence of the S-glycoprotein of the XBB.1 virus for mRNA synthesis was obtained. However, during the course of the experimental work, the composition of circulating strains changed, and by the time the combined vaccine was studied the most common variant was EG.5.1.1. This variant first appeared in March 2023, and had become dominant, acquiring three unique mutations in the spike protein relative to XBB.1 [45]. Therefore, the virus neutralization assay of mouse sera was conducted using these two virus variants. It was found that despite the mutations acquired by the Omicron variant EG.5.1.1 relative to XBB.1, the sera of mice vaccinated with both monovalent and combined mRNA formulations effectively neutralized it to the same extent. It is possible that these two variants of the Omicron lineage are not sufficiently different to detect a reduction in neutralizing antibody titers against the newer EG.5.1.1 variant. However, according to a study of the intranasal combined vaccine based on chimpanzee adenovirus AdC68-HATRBD, a decrease in neutralizing antibody titers against the evolutionarily latest variants was observed in the sera of vaccinated mice [41].

The immunological efficacy of the combined vaccine against influenza virus infection was tested in a mouse model and found to be 100% protective. Vaccination with the combined mRNA vaccine significantly reduced the viral load in the lungs of mice infected with both influenza virus and coronavirus. Thus, the combination of two mRNA-based vaccines does not affect the immunological efficacy of the influenza and coronavirus components in mice. Clearly, the ferret has proven to be a good model for studying influenza and appears to be the best mammalian model for immunological efficacy [46], but the lethal mouse model of influenza infection is widely used in such studies [43,44,47,48,49]. Mice receiving placebo in such studies died after influenza infection. In our research, the mice that received PBS instead of the vaccine all died on day 7. The use of mouse strains of influenza virus means that the mouse can still be used in studies of different aspects of the disease, and its small size and low cost allows researchers to conduct studies on a larger scale.

Hematological indicators of blood during coronavirus infection, according to clinical observations, may have prognostic value for determining the possible risks of severe disease progression in humans [50,51]. Therefore, we conducted blood studies in mice on the 4th and 7th days after coronavirus infection. In mouse models of coronavirus infection, the determination of hematological indicators is a rare subject in publications. Normally, nucleated erythroid cells are absent in peripheral blood. The presence of NRBCs in human blood is associated with poor outcomes in critically-ill patients with coronavirus infection [52]. In our work, NRBCs were reduced in the blood of mice in the mRNA-5 group compared to the PBS group, confirming the protective effect of the combined vaccine.

Our study highlights the advantages of the mRNA platform for creating combined vaccines as this delivery method provides the synthesis of native viral antigens directly in the host organism, and, therefore, has advantages over inactivated viruses [21]. Additionally, the lack of mutual influence on the immunogenicity levels of different components is a significant advantage. Dulfer et al. have shown that when people are simultaneously vaccinated with different types of vaccines (mRNA-based coronavirus vaccine and split inactivated influenza vaccine), lower immunogenicity, measured in binding antibody units (BAU), is observed compared to separate coronavirus vaccination [53]. However, the impact on the level of protection against symptomatic COVID-19 infection and severe disease was not studied. The study also did not examine the cellular and molecular determinants of the observed effect, but the authors suggest that such phenomena may be associated with the inhibition of the mRNA coronavirus vaccine due to the release of type I interferons (IFN I) induced by influenza vaccination. We speculate that our combined mRNA vaccine will have a chance to avoid such effect, due to the common platform for all components.

Our study has some limitations. One of these is that the antigenic composition of the studied vaccines is not fully up to date. However, this limitation is mitigated by the fact that, in this study, our goal was to demonstrate the principal effect. In the future, the development of vaccination protocols against these pathogens will require adherence to the concept of an annual update of the antigenic composition of vaccines. Another limitation is that we demonstrated efficacy only for one of the four influenza vaccine components (H1N1), which is related to the difficulties in adapting influenza viruses to the mouse animal model, especially for viruses from the H3N2 and B groups. Nevertheless, given the high HAI titer values, which for influenza viruses exceed the correlates of protection by more than 20 times [54], we believe that high protective activity can be expected due to the strong correlation between epidemiological efficacy and serum HAI titers in humans.

## 5. Conclusions

In this study, we investigated the feasibility of combining the mRNA vaccine against influenza and SARS-CoV-2. As for the conclusion, our combined mRNA vaccine has high immunogenicity and protective efficacy against influenza and SARS-CoV-2. Following the two-dose regimen of the combined vaccine, we demonstrated a high level of immunogenicity, which was not distinguishable from that of the monovalent vaccine for each component. The combination does not lead to a decrease in the virus-neutralizing activity of the sera of vaccinated animals against the two Omicron strains. The combined mRNA vaccine protected mice from death in viral challenge experiments, providing 100% survival when subjected to influenza challenge and an absence of the viral load in lungs in coronavirus challenge. We believe that such a combined mRNA-based vaccine could be a good alternative to separate human vaccinations for the prevention of COVID-19 and influenza.

## Figures and Tables

**Figure 1 vaccines-12-01206-f001:**
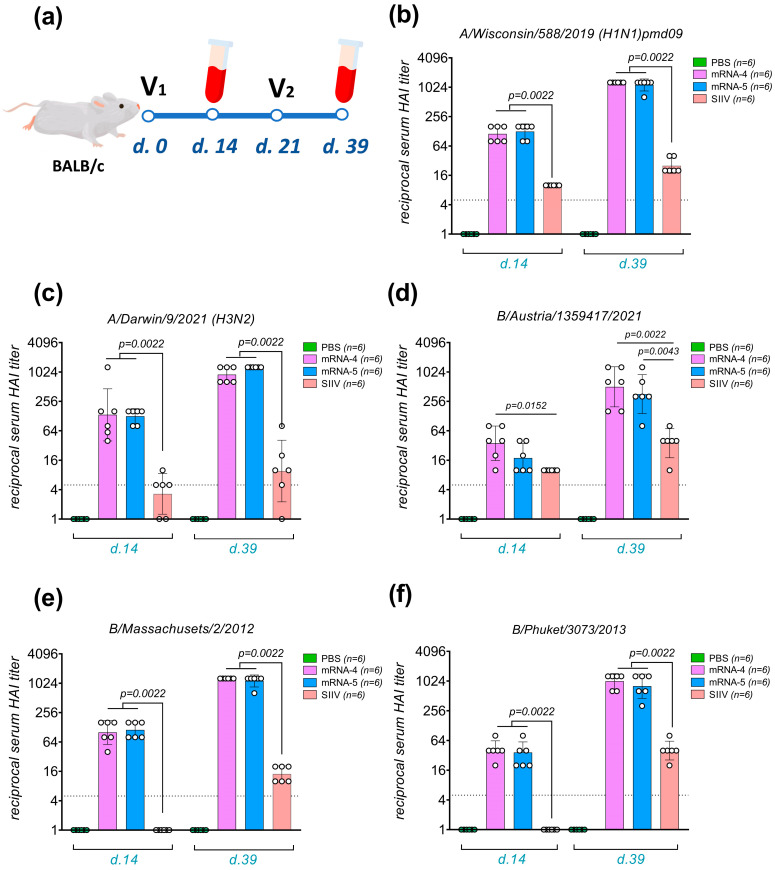
Study of the immunogenicity of influenza components of the combined vaccine. Mice immunized with multivalent mRNA vaccines exhibit a strong humoral immune response against the influenza virus antigens. (**a**)—the design of experiment for determining the immunogenicity levels of the vaccine formulations under study. HAI titers in serum were measured 14 days after the first vaccination (V1) and 18 days after the second vaccination (V2) using antigens of influenza virus strains homologous to those in the vaccines; (**b**)—A/Wisconsin/588/2019 (H1N1)pmd09; (**c**)—A/Darwin/9/2021 (H3N2); (**d**)—B/Austria/1359417/2021 (B/Victoria lineage); (**e**)—B/Massachusetts/02/2012 (B/Yamagata lineage); and (**f**)—B/Phuket/3073/2013. mRNA-5 refers to the combined RNA-LNP containing mRNAs encoding the hemagglutinins of four influenza virus strains and the coronavirus spike protein; mRNA-4 refers to the combined mRNA-LNP containing mRNAs encoding the hemagglutinins of four influenza virus strains; SIIV—split inactivated vaccine; and PBS—phosphate-buffered saline. The data are representative of a single experiment and are presented as geometric means with standard deviations from the geometric mean. The circles show the individual value. Statistical comparisons were conducted using the Mann–Whitney test.

**Figure 2 vaccines-12-01206-f002:**
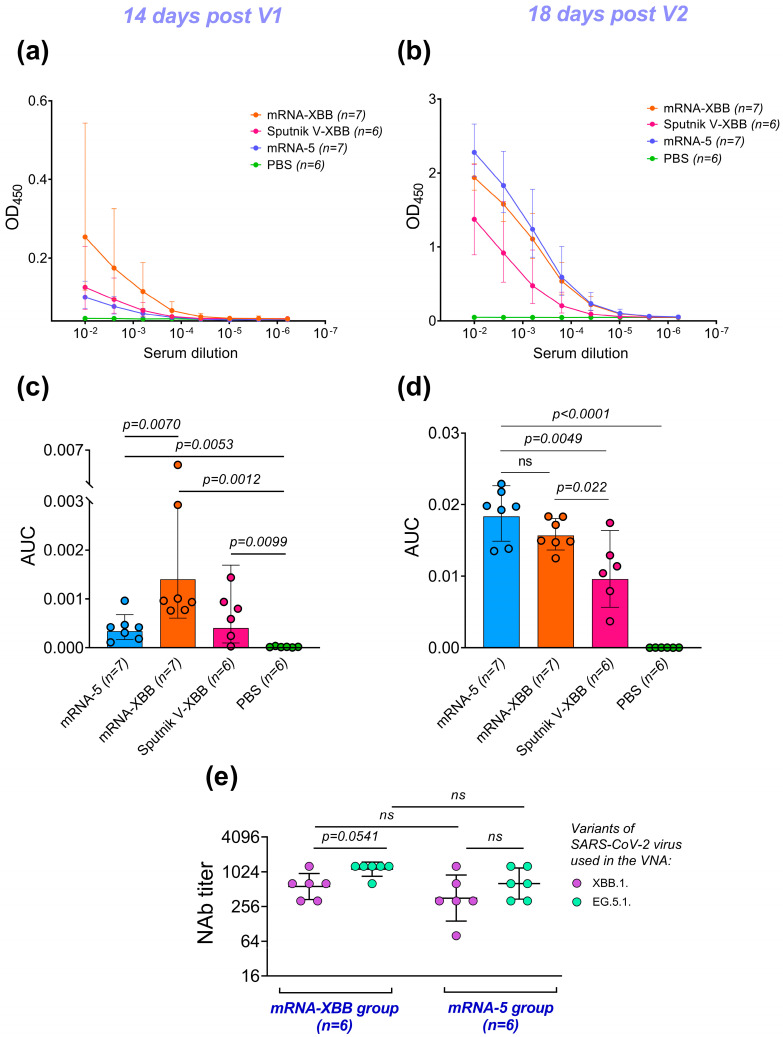
Immunogenicity of the SARS-CoV-2 S-glycoprotein in various compositions. Vaccination of mice with multivalent mRNA vaccines induces a strong humoral immune response to the SARS-CoV-2 spike protein (XBB.1). (**a**) and (**b**) show ELISA curves representing the geometric mean OD_450_ values at the corresponding serum dilutions of vaccinated animals after the first and second vaccinations, respectively. (**c**) and (**d**) show a comparison of the area under the curve (AUC) values for the ELISA curves after the first and second vaccinations, respectively. (**e**) shows the virus-neutralizing activity of mouse serum on day 39 of the experiment. mRNA-5 refers to the combined RNA-LNP containing mRNAs encoding the hemagglutinins of four influenza virus strains and the coronavirus spike protein; mRNA-XBB refers to the mRNA-LNP formulation containing only the coronavirus spike protein mRNA; Sputnik V-XBB—approved updated vector-based vaccine; and PBS—phosphate-buffered saline. The circles show the individual value. Comparisons between groups immunized with different vaccines were conducted using the Mann–Whitney test and unpaired *t*-test (for comparisons between the mRNA-XBB group and others), while comparisons of titers against the two coronavirus strains were performed using the Wilcoxon rank-sum test (ns - not significant).

**Figure 3 vaccines-12-01206-f003:**
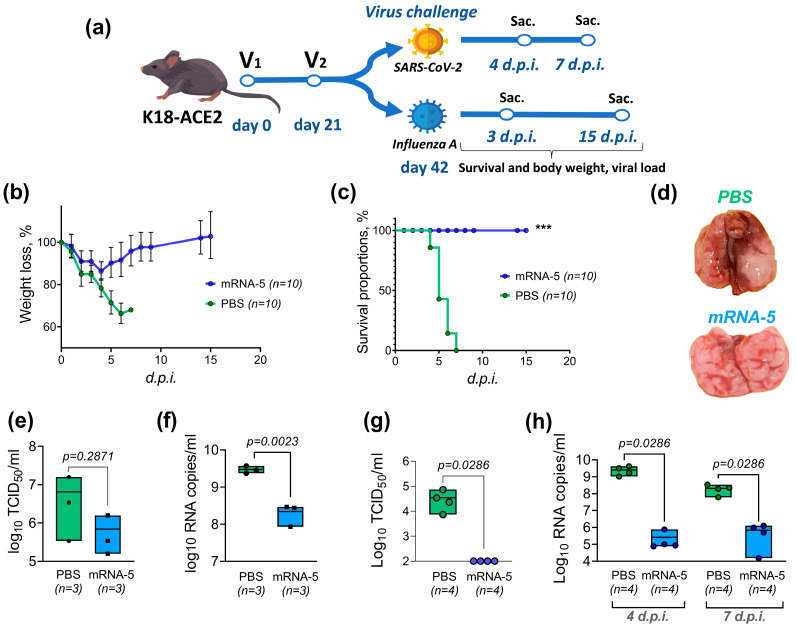
Efficacy of the combined mRNA vaccine. The combined mRNA vaccine protects mice from lethal outcomes and high viral load during separate infections with influenza A/Victoria/2570/2019 (H1N1)pmd09 virus and SARS-CoV-2 virus. (**a**)—a flowchart of experiments investigating the efficacy of mRNA-5 in separate influenza and coronavirus infections. Influenza A virus challenge results are presented in the weight loss dynamics (**b**) and survival rates after infection (**c**), the photos of lungs isolated from mice (**d**), the amount of virus (**e**), and the viral genome copies (**f**) in the lungs of animals on the 3rd day after infection. SARS-CoV-2 virus results are presented in the amount of SARS-CoV-2 virus on the 4th day post infection (**g**) and viral genome copies in the lungs of animals on the 4th and 7th days after infection (**h**). Survival data were compared using the Mantel–Cox test. ***—significance criterion *p* ≤ 0.001. mRNA-5—combined mRNA vaccines and PBS—phosphate-buffered saline solution.

**Figure 4 vaccines-12-01206-f004:**
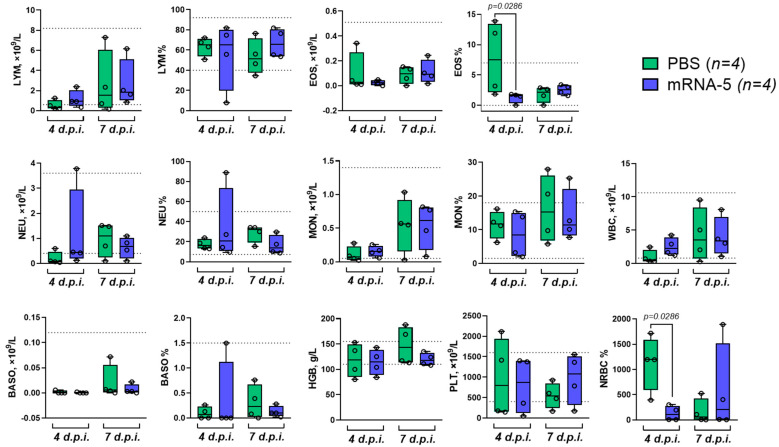
Hematological studies of mice vaccinated with mRNA-5 and infected with SARS-CoV-2 virus. Whole blood samples from mice were examined on the 4th and 7th days after coronavirus infection in experiments assessing the efficacy of the mRNA-5 vaccine on K18-hACE2 mice. Comparisons between groups were made using the nonparametric Mann–Whitney test. mRNA-5 is a combined mRNA vaccine, and PBS is a phosphate-buffered saline solution. The dotted lines indicate the range of normal Smart V5 Vet laboratory values for mice. The circles show the individual value. LYM—lymphocytes, EOS—eosinophils, NEU—neutrophils, MON—monocytes, WBC—white blood cells, BASO—basophils, HGB—hemoglobin, PLT—platelet count, NRBC—nuclear red blood cells, and d.p.i.—days post infection.

**Table 1 vaccines-12-01206-t001:** Viral composition of the vaccines tested.

mRNA-5	mRNA-4	SIIV	mRNA-XBB	Sputnik V
A/Wisconsin/588/2019 (H1N1)pmd09	A/Wisconsin/588/2019 (H1N1)pmd09	A/Victoria/2570/2019 (H1N1)pdm09	SARS-CoV-2 XBB.1	SARS-CoV-2 XBB.1
A/Darwin/9/2021 (H3N2)	A/Darwin/9/2021 (H3N2)	A/Darwin/9/2021 (H3N2)		
B/Austria/1359417/2021 (B/Victoria lineage)	B/Austria/1359417/2021 (B/Victoria lineage)	B/Austria/1359417/2021 (B/Victoria lineage)		
B/Massachusetts/02/12 (B/Yamagata lineage)	B/Massachusetts/02/12 (B/Yamagata lineage)	B/Phuket/3073/2013 (B/Yamagata lineage)		
SARS-CoV-2 XBB.1				

## Data Availability

The data presented in this study are available on request from the corresponding author.

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
