# Peer review of "Immunogenicity and Efficacy of Combined mRNA Vaccine Against Influenza and SARS-CoV-2 in Mice Animal Models"

_vaccines, 2024, doi:10.3390/vaccines12111206_

Round 1

Reviewer 1 Report

Comments and Suggestions for Authors

In this manuscript, the authors aimed to test immunogenicity and efficacy of a multivalent mRNA vaccine in a pre-clinical animal model. The combined vaccine (mRNA-5) that encodes five antigens was used to vaccine mice using primer and boost approach where booster dose was administered at 21 days after first dose. Immunogenicity of the vaccine was confirmed using in vitro antibody binding and virus neutralization assays. The multivalent vaccine induced robust protection in mouse model evident from reduced virus replication and improved survival upon virus challenge. Most of the claims are sufficiently justified by data presented. However, some improvements are required prior to publication, these are listed below.

·   Figure panel are annotated in a non-standard fashion, which is very confusing. This should be changed to standard approach, i.e. panel label on top right side.

·  The sample number and number of repeats are missing from some figure legends, this should be described in all legends for the sake of transparency.

·     In Figure 2 (a), it is unclear why the mRNA-5 vaccine did not result in similar OD450 values to the mRNA-XBB at 14 days post V1. Can the authors please explain this discussion section.

·   In Figure 3, the lungs from the PBS and mRNA-5 groups are displayed differently (lobes clumped versus spread out, respectively). If there are additional and/or better representative images, can the authors please show them instead. Further, if the number of foci per lobe in each vaccine group have been quantified, can this data be added in the same Figure.

·     Figure legend 3 states: “…Influenza A virus challenge results are presented in the survival rates (b) and weight loss dynamics after infection (c)…”. These figure sub-letters seems to be switched, please correct.

·    In Figure 4, the legend for this whole blood data figure makes reference to lung data collected in Figure 3, making it unclear as to which samples were analysed. Please rephrase/modify the language.

·   On page 13, line 479, the authors claim “…. In our work NRBCs were absent in blood of mice …”. This is not correct as NRBCs can still be detected in some mRNA-5-immunised mice both 4 and 7 days post-infection. Please correct the text (e.g. “…NRBCs were reduced…”).

Comments on the Quality of English Language

Inconsistent colour schemes and terminologies are used throughout the manuscript. For instance, “d. 39” and “18 days post-V2” in Figure 1 and “. mRNA-5 vs 5-mRNA”. I would recommend to use a consistent terminology and colour schemes for the figures. throughout the manuscript. Moreover, the grammar and readability can be improved.

Author Response

Comments 1: Figure panel are annotated in a non-standard fashion, which is very confusing. This should be changed to standard approach, i.e. panel label on top right side.

Response 1: Thank you for your thorough review and valuable suggestions. Regarding the first comment, thank you for bringing this to our attention. We regret that the style of figure annotation used made the manuscript difficult to read. However, we used the Microsoft Word template that shows the following example of figure annotation when preparing the manuscript: https://www.mdpi.com/files/word-templates/vaccines-template.dot

And we used the recommended by template style. Based on previously published articles in the Vaccines, we found that the most popular style for labelling a figure in a panel is top left. But you recommend changing to top right. Is there a mistake here? Perhaps you meant the upper left? We have modified the style of figures annotation and labeled all figures in the panel on top left side.

Comments 2: The sample number and number of repeats are missing from some figure legends, this should be described in all legends for the sake of transparency.

Response 2: Agree. We have, accordingly, modified Figures 1-3. In Figure 1, we have added the sample number to the figure legends for each figure in the panel. In Figure 2, we have changed the color of graphs, added the sample number to the legends for each figure in the panel and p-value for comparison of mRNA-5 and mRNA-XBB groups, which was missed in the first version of manuscript. In Figure 3, we have added the sample to the figure legends.

Comments 3: In Figure 2 (a), it is unclear why the mRNA-5 vaccine did not result in similar OD450 values to the mRNA-XBB at 14 days post V1. Can the authors please explain this discussion section.

Response 3: Thank you for pointing this out. We agree with this comment. Therefore, we have added a discussion of the effect obtained on lines 436-452:

“Regarding the immunogenicity of the coronavirus component of the vaccine, it is worth noting that the combined mRNA vaccine demonstrates high immunogenicity, indistinguishable from that of the monovalent vaccine after second vaccination. However, after first vaccination significant excess of immunogenicity level in mice received monovalent mRNA vaccine compared to multivalent was observed. We can speculate that the effect obtained is related to the specificity of the primary immune response development, which is based on the ability of local professional antigen-presenting cells to take up antigen and present it to T-cells and non-specific effector cells. Although mice in the mRNA-5 and mRNA-XBB groups received the same amount of mRNA encoding the spike antigen, mice in the multivalent group received a total of 5 times more mRNA (25 µg per mouse). Therefore, mice from these groups received different amounts of antigen load per conditionally equal number of regional DCs, which could be the result of a decrease in the level of binding antibodies to the coronavirus adhesion protein. A similar effect of mutual influence of vaccine mRNA components on the level of immunogenicity has been previously demonstrated [20,41]. In two other studies, vaccination with the combination of 4 to 20 mRNAs encoding influenza antigens did not result in reduced immunogenicity compared to monovalent preparations [42,43]. However, the dose of monovalent mRNAs and combinations per mouse, the route of administration, time of sera collection were different in our study, so a direct comparison is not relevant.”

Comments 4: In Figure 3, the lungs from the PBS and mRNA-5 groups are displayed differently (lobes clumped versus spread out, respectively). If there are additional and/or better representative images, can the authors please show them instead. Further, if the number of foci per lobe in each vaccine group have been quantified, can this data be added in the same Figure.

Response 4: Thank you for pointing this out. We agree. We have additional image of the lungs from mRNA-5 group with clumped lobes. Figure 3 d was accordingly changed, but the number of foci per lobe haven’t been quantified.

Comments 5: Figure legend 3 states: “…Influenza A virus challenge results are presented in the survival rates (b) and weight loss dynamics after infection (c)…”. These figure sub-letters seems to be switched, please correct.

Response 5: Thanks for pointing out the mistake. We have corrected Figure legend 3 on lines 338-343.

Comments 6: In Figure 4, the legend for this whole blood data figure makes reference to lung data collected in Figure 3, making it unclear as to which samples were analysed. Please rephrase/modify the language.

Response 6: We agree, it was unclear. We have changed the legend to make it easier to understand, deleting the description of manipulation of lung isolation on lines 378-385:

“Hematological studies of mice mRNA-5 vaccinated and infected with SARS-CoV-2 virus. Whole blood samples from mice were examined on the 4th and 7th days after coronavirus infection in experiments assessing the efficacy of the mRNA-5 vaccine on K18-hACE2 mice. Comparisons between groups were made using the nonparametric Mann-Whitney test. mRNA-5 is a combined mRNA vaccine, and PBS is a phosphate-buffered saline solution. The dotted lines indicate the range of normal Smart V5 Vet laboratory values for mice. LYM – lymphocytes, EOS – eosinophils, NEU – neutrophils, MON – monocytes, WBC – white blood cells, BASO – basophils, HGB – hemoglobin, PLT – platelet count, NRBC – nuclear red blood cells, d.p.i. – days post infection.”

Comments 7: On page 13, line 479, the authors claim “…. In our work NRBCs were absent in blood of mice …”. This is not correct as NRBCs can still be detected in some mRNA-5-immunised mice both 4 and 7 days post-infection. Please correct the text (e.g. “…NRBCs were reduced…”).

Response 7: Thank you, that's a fair point. We have accordingly modified the text on lines 489-490: “In our work, NRBCs were reduced in the blood of mice in the mRNA-5 group compared to the PBS group, confirming the protective effect of combined vaccine.”

4. Response to Comments on the Quality of English Language

Point 1: Inconsistent colour schemes and terminologies are used throughout the manuscript. For instance, “d. 39” and “18 days post-V2” in Figure 1 and “. mRNA-5 vs 5-mRNA”. I would recommend to use a consistent terminology and colour schemes for the figures. throughout the manuscript. Moreover, the grammar and readability can be improved.

Response 1: Thank you for pointing this out. We have brought the color of the graphs and terminology into a consistent style.

Reviewer 2 Report

Comments and Suggestions for Authors

The Development of a combined mRNA vaccine that targets both Influenza virus and SARS-CoV-2 are important to against the viruses infection. In the present study, the authors tested the combined vaccine 2 against Influenza and SARS-CoV-2 in mice models based on mRNA technology. The results showed that the combined vaccine provided strong protection for the mice against infection by either the influenza virus or SARS-CoV-2. While the use of a combined vaccine is an exciting approach, further improvements are still needed.

Major points:

1. The effectively of the combined vaccine on elicits an antigen-specific T cell immune response should be checked.

2. As combined vaccine including the HA from influenza B virus, after immunization, the combined vaccine should be also challenged by influenza B virus.

3. More experiments should be done to test the efficacy of the combined mRNA vaccine, such as viruses RNA copy, histopathological analysis from tissue samples.

Minor Points:

1. The HA expression level of each component of influenza virus HA mRNA in the vaccine should be checked in the mRNA-4.

2. The HA specific IgG should be tested by ELISA.

3. The RBD expression level of SARS-CoV-2 in the mRNA-5 should be checked.

4. The RBD specific IgG should be tested by ELISA.

5. The mice model used for the combined vaccine was K18- 335 ACE2, please show that if the mice is a good model for influenza virus infection.

6. It is better to compare the cytokines and chemokines levels after infection between vaccinated mice and control mice.

7. The abstract should be improved to highlight the conclusions, and clear the background.

8. The conclusion of each result part should be clear.

9. The References part should be checked. Such as 27, 38, 39.

Comments on the Quality of English Language

The English writing should be improved, such as the description should be clearer.

Author Response

Comments 1: The effectively of the combined vaccine on elicits an antigen-specific T cell immune response should be checked.

Response 1: Thank you for your thorough review and valuable suggestions. Evaluation of the T-cell immune response induced by combined mRNA vaccine is certainly an important task. However, the main objectives of the work were to study the humoral immune response as sufficient correlate of protection for these infections (SARS-CoV-2 and Influenza virus) and the protective efficacy per se. The humoral immune response is crucial for protection against these viral infections and is the basic defense mechanism for a successful vaccine. In addition, neutralizing antibody titers and HAI titers are proven and recognized correlates of protection for coronavirus and influenza respectively, in contrast to the T-cell response.

Comments 2: As combined vaccine including the HA from influenza B virus, after immunization, the combined vaccine should be also challenged by influenza B virus.

Response 2: This is a fair point. Efficacy in the non-clinical study report must be confirmed for all five components of the combination vaccine. And we plan to do this in non-clinical studies of the vaccine, but this requires a special animal model. There are great difficulties with adapting the influenza B virus to mice, which requires a transition to using ferrets instead of laboratory mice. Ferrets are the gold standard for animal studies of vaccines and drugs against the influenza virus, but they are dangerous predators, working with which requires different conditions and is very expensive. In this regard, we will be able to conduct such experiments only at the next stage of our research when we receive funding for full-fledged preclinical studies.

Comments 3: More experiments should be done to test the efficacy of the combined mRNA vaccine, such as viruses RNA copy, histopathological analysis from tissue samples.

Response 3: Thank you for pointing this out. The design of the animal studies was guided by the 3Rs. Our manuscript includes the results of viral RNA copy assays, macroscopic pathological observations of the lungs. We emphasize in the manuscript that the results reproduced those previously obtained for individual mRNA vaccines, further confirming the absence of chance in the phenomena observed.

Comments 4: The HA expression level of each component of influenza virus HA mRNA in the vaccine should be checked in the mRNA-4.

Response 4: Thank you for the comment. We confirmed the functional activity of mRNA encoding HA (H1N1) or Spike SARS-CoV-2 in two previous studies using immunocytochemical staining and ELISA of transfected cell lysates, respectively. Therefore, we have added references to these results on lines 417-418 in the text.

Comments 5: The HA specific IgG should be tested by ELISA.

Response 5: Thank you for pointing this out. However, we chose HAI because it is the gold standard for assessing the immunogenicity of influenza vaccines and the efficacy of immunity against novel virus variants. In addition, in our previous study, HAI and ELISA results for different haemagglutinin domains correlated well. In the actual work, we chose to use the HAI method because it reflects the quantity of neutralizing antibodies, which play a key role in the first antiviral defense.

Comments 6: The RBD specific IgG should be tested by ELISA.

Response 6: Agree. Manuscript already contains the estimation of RBD specific IgG tested by ELISA. The results are shown in the section “3.2.2. Immunogenicity of the SARS-CoV-2 S-glycoprotein” on lines 287-294, 319-325 and in the Figure 2 a-d.

Comments 7: The mice model used for the combined vaccine was K18- 335 ACE2, please show that if the mice is a good model for influenza virus infection.

Response 7: Thank you for pointing this out. We have, accordingly, added a rationale for the relevance of the mouse model in the discussion section to emphasize this point on lines 474-481:

“Clearly, the ferret has proven to be a good model for studying influenza and appears to be the best mammalian model for immunological efficacy [45], but the lethal mouse model of influenza infection is widely used in such studies [43,46–49]. Mice receiving placebo in such studies died after influenza infection. In our research, the mice that received PBS instead of the vaccine all died on day 7. The use of mouse strains of influenza virus means that the mouse can still be used in studies of different aspects of the disease, and its small size and low cost allows researchers to conduct studies on a larger scale.”

Comments 8: It is better to compare the cytokines and chemokines levels after infection between vaccinated mice and control mice.

Response 8: Thank you for your comments. Analysis of the cytokines and chemokines levels requires preliminary studies to select sample preparation conditions, sampling time and sample type (whole blood, lavage, lung homogenates). This is important because exposures such as vaccination and infection can each lead to changes in the levels of cytokines and chemokines on their own. Our goal was to test the possibility of combination without significant reduction of immunogenicity for each mRNA vaccine component, while maintaining the protective effect against lethal infection.

Comments 9: The abstract should be improved to highlight the conclusions, and clear the background.

Response 9: Agree. We have, accordingly, revised the abstract to emphasize this point on lines 26-42. Now it looks like this:

“The combined or multivalent vaccines are actively used in pediatric practice and offer a series of advantages, including reduced number of injections and visits to the doctor, simplicity of the vaccination schedule and minimization of side effects, easier vaccine monitoring and storage, and lower vaccination costs. The practice of widespread use of the combined vaccines has shown the potential to increase vaccination coverage against single infections. mRNA platform has been shown to be effective against the COVID-19 pandemic and enables the development of combined vaccines. There are currently no mRNA-based combined vaccines approved for use in human. Some studies have shown that different mRNA components in a vaccine can interact to increase or decrease the immunogenicity and efficacy of the combined vaccine. In the present study, we investigated the possibility of combining the mRNA vaccines encoding seasonal influenza and SARS-CoV-2 antigens. In our previous works, both vaccine candidates have shown excellent immunogenicity and efficacy profiles in mice. In this work we demonstrated that the individual mRNA components of the combined vaccine did not affect the immunogenicity level of each other. The combined vaccine demonstrated excellent protective efficacy providing 100% survival when mice were infected with H1N1 influenza virus and reducing the viral load in the lungs. Four days after challenge with SARS-CoV-2 EG.5.1.1., no viable virus and low levels of detectable viral RNA were observed in the lungs of vaccinated mice.”

Comments 10: The conclusion of each result part should be clear.

Response 10: Agree. We have, changed the conclusion, according with your comments on lines 516-527. Now it looks like this:

“In this study, we investigated the feasibility of combining mRNA vaccine against influenza and SARS-CoV-2. As for conclusion, our combined mRNA vaccine has high immunogenicity and protective efficacy against influenza and SARS-CoV-2. Following the two-dose regimen of the combined vaccine, we demonstrated a high level of immunogenicity, not distinguishable from that of the monovalent vaccine for each component. The combination does not lead to a decrease in the virus-neutralizing activity of the sera of vaccinated animals against the two Omicron strains. The combined mRNA vaccine protected mice from death in viral challenge experiments, providing 100% survival in influenza challenge and absence of the viral load in lungs in coronavirus challenge. We believe that such combined mRNA-based vaccine could be a good alternative to separate human vaccination for the prevention of COVID-19 and influenza”

Comments 11: The References part should be checked. Such as 27, 38, 39.

Response 11: Agree. We have corrected these references, translating it to English. Now it looks like this:

“27.  On Approval of Sanitary Rules and Norms SanPiN 1.2.3685-21 “Hygienic Standards and Requirements for Ensuring Safety and (or) Harmlessness of Environmental Factors for Humans” Dated January 28, 2021 - Docs.Cntd.Ru Available online: https://docs.cntd.ru/document/573500115 (accessed on 7 October 2024).

38.        COVID-19 Vaccination in Russia. Wikipedia 2024.

39.  The Ministry of Health has registered the updated Sputnik V vaccine Available online: https://www.kommersant.ru/doc/6535351 (accessed on 22 August 2024).”

4. Response to Comments on the Quality of English Language

Point 1: The English writing should be improved, such as the description should be clearer.

Response 1: Thank you for the pointing this out. We have edited the text, especially in the parts where it was difficult to read, and simplified some sentences.

Reviewer 3 Report

Comments and Suggestions for Authors

The manuscript submitted by Mazunina et al., reported the immunogenicity and efficacy of combined mRNA vaccine against Influenza and SARS-CoV-2 in mice animal models. In the manuscript, the combined mRNAs vaccine encoding seasonal influenza and coronavirus antigens exhibited ideal immune protection effects with no negative mutual effect in mice immunization.  The data in this manuscript has certain reference value. However, the manuscript has obvious shortcomings.

1, in the abstract, the content of abstract is unreasonable, which should highlight the research results and significance.

2. The result lacks innovation. Because the author has previously confirmed that two individual vaccines have good immune effects. So we can speculate that the combined vaccines may have a similar effect. How to demonstrate the innovation of this data in this manuscript?

3.Figure 3d, there is lack proportional size.

4.In Line 385,  4th should be revised to 4th.

5.In animal experiments in Figure 4, PBS control is not suitable and mRNA vector control should be used instead.

6. In the discussion, too many reported results were introduced. The data in the manuscript should be combined with the confirmed results for analysis and discussion in section of discussion.

Author Response

Comments 1: In the abstract, the content of abstract is unreasonable, which should highlight the research results and significance.

Response 1: Thank you for pointing this out. We have modified the Abstract on lines 26-42:

“The combined or multivalent vaccines are actively used in pediatric practice and offer a series of advantages, including reduced number of injections and visits to the doctor, simplicity of the vaccination schedule and minimization of side effects, easier vaccine monitoring and storage, and lower vaccination costs. The practice of widespread use of the combined vaccines has shown the potential to increase vaccination coverage against single infections. mRNA platform has been shown to be effective against the COVID-19 pandemic and enables the development of combined vaccines. There are currently no mRNA-based combined vaccines approved for use in human. Some studies have shown that different mRNA components in a vaccine can interact to increase or decrease the immunogenicity and efficacy of the combined vaccine. In the present study, we investigated the possibility of combining the mRNA vaccines encoding seasonal influenza and SARS-CoV-2 antigens. In our previous works, both vaccine candidates have shown excellent immunogenicity and efficacy profiles in mice. In this work we demonstrated that the individual mRNA components of the combined vaccine did not affect the immunogenicity level of each other. The combined vaccine demonstrated excellent protective efficacy providing 100% survival when mice were infected with H1N1 influenza virus and reducing the viral load in the lungs. Four days after challenge with SARS-CoV-2 EG.5.1.1., no viable virus and low levels of detectable viral RNA were observed in the lungs of vaccinated mice.”

Comments 2: The result lacks innovation. Because the author has previously confirmed that two individual vaccines have good immune effects. So we can speculate that the combined vaccines may have a similar effect. How to demonstrate the innovation of this data in this manuscript?

Response 2: Thank you for your comment, we have highlighted the discussion of this issue in the manuscript at the lines 437-453 and 489-490. Unfortunately, in order to claim that combining does not affect the immunogenicity of each component, it is necessary to test this in animal experiments. So, we describe it in the current manuscript. Mutual effects of components in combined mRNA vaccines are not uncommon. For example, such effects have been shown in earlier animal studies [Hajnik, R.L.; Plante, J.A.; Liang, Y.; Alameh, M.-G.; Tang, J.; Bonam, S.R.; Zhong, C.; Adam, A.; Scharton, D.; Rafael, G.H.; et al. Dual Spike and Nucleocapsid mRNA Vaccination Confer Protection against SARS-CoV-2 Omicron and Delta Variants in Preclinical Models. Sci. Transl. Med. 2022, 14, eabq1945, doi:10.1126/scitranslmed.abq1945; Bykonia, E.N.; Kleymenov, D.A.; Gushchin, V.A.; Siniavin, A.E.; Mazunina, E.P.; Kozlova, S.R.; Zolotar, A.N.; Usachev, E.V.; Kuznetsova, N.A.; Shidlovskaya, E.V.; et al. Major Role of S-Glycoprotein in Providing Immunogenicity and Protective Immunity in mRNA Lipid Nanoparticle Vaccines Based on SARS-CoV-2 Structural Proteins. Vaccines 2024, 12, 379, doi:10.3390/vaccines12040379;]. There is also a study on the effect of simultaneous vaccination of human against influenza and coronavirus with two different vaccines. Researchers showed reduced immunogenicity of the coronavirus vaccine after using such regimen of vaccination [Dulfer, E.A.; Geckin, B.; Taks, E.J.M.; GeurtsvanKessel, C.H.; Dijkstra, H.; Van Emst, L.; Van Der Gaast – De Jongh, C.E.; Van Mourik, D.; Koopmans, P.C.; Domínguez-Andrés, J.; et al. Timing and Sequence of Vaccination against COVID-19 and Influenza (TACTIC): A Single-Blind, Placebo-Controlled Randomized Clinical Trial. Lancet Reg. Health - Eur. 2023, 29, 100628, doi:10.1016/j.lanepe.2023.100628.]. We have cited these studies in the discussion section of the manuscript.

We believe that the publication of experimental results is important even if it partially repeats previously obtained results, since based on multiple reports of the same effects shown in different laboratories, one can judge the reliability of the results obtained and their reproducibility. As we can see from the data presented in our article, some effects are reproducible, and some are not. We think that reproducibility of results is especially important for developments that become drugs, since the better the effects are studied and described, the lower the risk that we will encounter an unexpected negative effect when translating the results into practice. In this regard, our study does not completely repeat any study, in some ways it coincides, and in some ways, it diverges from other already published studies.

Comments 3: Figure 3d, there is lack proportional size.

Response 3: Thank you for pointing this out. We have changed the size of photos in the Figure 3.

Comments 4: In Line 385, 4th should be revised to “4th”.

Response 4: Thank you for pointing this out. We have corrected it through manuscript.

Comments 5: In animal experiments in Figure 4, PBS control is not suitable and mRNA vector control should be used instead.

Response 5: Perhaps the legend to the Figure 4 does not fully capture the idea of the experiment. We have corrected it on lines 378-385. The aim of this experiment was to evaluate the effect of vaccination of animals on blood parameters after their infection with coronavirus. Based on the 3R principles, blood samples of animals obtained in the efficacy study were examined for this purpose. We believe that mice receiving placebo - PBS as a control group was appropriate for this aim.

Comments 6: In the discussion, too many reported results were introduced. The data in the manuscript should be combined with the confirmed results for analysis and discussion in section of discussion.

Response 6: Agree, in particular, by accident we have duplicated the additional description of the results in the discussion section on immunological effectiveness. We have corrected this section.

Round 2

Reviewer 2 Report

Comments and Suggestions for Authors

The revised version has been greatly improved. It would be even better if the scientific soundness could be enhanced in the abstract.

Author Response

Comments 1: The revised version has been greatly improved. It would be even better if the scientific soundness could be enhanced in the abstract.

Response 1: Thank you very much for taking the time to review this manuscript. Your comments have helped improve the manuscript.

Reviewer 3 Report

Comments and Suggestions for Authors

In the manuscript resubmitted by Elena P. Mazunina, the author has made reasonable revisions to the suggestions made by the reviewers. Also, the author provided reasonable explanations for some of the content in the revised manuscript. I think this data has certain reference value, and I suggest accepting this manuscript.

Author Response

Comments 1: In the manuscript resubmitted by Elena P. Mazunina, the author has made reasonable revisions to the suggestions made by the reviewers. Also, the author provided reasonable explanations for some of the content in the revised manuscript. I think this data has certain reference value, and I suggest accepting this manuscript.

Response 1: Thank you for taking the time to review this manuscript. Your comments have been very useful in improving the manuscript.